# Poly(Ethylene Furanoate) along Its Life-Cycle from a Polycondensation Approach to High-Performance Yarn and Its Recyclate

**DOI:** 10.3390/ma14041044

**Published:** 2021-02-23

**Authors:** Tim Höhnemann, Mark Steinmann, Stefan Schindler, Martin Hoss, Simon König, Antje Ota, Martin Dauner, Michael R. Buchmeiser

**Affiliations:** German Institutes of Textile and Fiber Research, Koerschtalstr. 26, D-73770 Denkendorf, Germany; tim.hoehnemann@ditf.de (T.H.); stefan.schindler@ditf.de (S.S.); martin.hoss@ditf.de (M.H.); simon.koenig@ditf.de (S.K.); antje.ota@ditf.de (A.O.); michael.buchmeiser@ditf.de (M.R.B.)

**Keywords:** polycondensation, melt, solid-state, 2,5-furan dicarboxylic acid, polyethylene furanoate, fiber spinning, yarn drawing, crystallinity

## Abstract

We report on the pilot scale synthesis and melt spinning of poly(ethylene furanoate) (PEF), a promising bio-based fiber polymer that can heave mechanical properties in the range of commercial poly(ethylene terephthalate) (PET) fibers. Catalyst optimization and solid state polycondensation (SSP) allowed for intrinsic viscosities of PEF of up to 0.85 dL·g^−1^. Melt-spun multifilament yarns reached a tensile strength of up to 65 cN·tex^−1^ with an elongation of 6% and a modulus of 1370 cN·tex^−1^. The crystallization behavior of PEF was investigated by differential scanning calorimetry (DSC) and XRD after each process step, i.e., after polymerization, SSP, melt spinning, drawing, and recycling. After SSP, the previously amorphous polymer showed a crystallinity of 47%, which was in accordance with literature. The corresponding XRD diffractograms showed signals attributable to α-PEF. Additional, clearly assignable signals at 2θ > 30° are discussed. A completely amorphous structure was observed by XRD for as-spun yarns, while a crystalline phase was detected on drawn yarns; however, it was less pronounced than for the granules and independent of the winding speed.

## 1. Introduction

Sustainability is a fundamental challenge for the polymer industry and, especially, for the textile industry. A promising solution is the substitution of fossil fuel-based plastics by bioplastics. According to IUPAC, the term “bioplastics” stands for “bio-based poly-mers” and refers to polymers on the basis of “biomass or monomers derived from biomass […]” [1]. FDCA (2,5-furan dicarboxylic acid) is considered to be one of the most important platform chemicals of the bio-based chemical industry due to its chemical and structural similarity to the widely used and well-established but petrochemically-produced terephthalic acid (TPA), from which polyethylene terephthalate (PET) is synthesized. In this context, FDCA-based polyethylene furanoate (PEF) represents a possible bio-based fiber polymer. When the synthetic route employing mono-ethylene glycol (MEG) is used, it can be 100% bio-based. Production chains for FDCA and its precursor 5-HMF (5-hydroxymethylfurfural) from waste biomass have already been shown and are outside the food chain [2,3,4,5,6]. Though PEF is not biodegradable at the current state of research, it is fully recyclable [7] and has several advantages over PET, such as a glass transition temperature *T_g_* that is almost 15 °C above that of PET (85–89 °C vs. ~70 °C) [8,9,10] and a melting range that is about 40 °C lower (~210–215 °C for PEF compared to 250–260 °C for PET) [8,9,11,12,13]. Thus, though the processing window for texturizing and drawing is narrower than that of PET, it can be used at higher service temperatures and requires less conversion energy in melt processing.

For the final fiber properties, the orientation of the polymer chains is a major driver. Generally, the production of high-strength yarns takes places in a two-stage process and comprises the spinning and subsequent drawing of the yarn on a draw twisting or a draw winding machine [14]. As-spun yarns are referred to as LOY (low oriented yarn) or POY (partially oriented yarn), depending on the winding speed, which is <1200 m·min^−1^ for LOY [14] and between 2000 and 4200 m·min^−1^ for POY [15]. Both, LOY and POY can be converted into FDY (fully drawn yarn) by hot drawing. In course of this process, the final properties, especially the titer and the mechanics, are adjusted by optimizing both the orientation of the polymer chains and the crystallinity [16]. The strength of a yarn increases with stretching, which inevitably reduces the maximum elongation unless further measures are taken [17] and the maximum draw ratio of the yarn is limited. Common draw ratios range from 3.0 to 6.0 for LOY and from 1.5 to 2.0 for POY [18]. In general, higher strengths can be achieved with LOY due to its better orientation properties, but LOY has a limited storage time. LOY must not be stored over a long time, because progressive crystallization can lead to embrittlement [19,20,21,22,23], which is usually not the case for POY [24]. 

The crystallization behavior and, especially, the investigation of the crystal structure of PEF by XRD have already been studied [8,9,10,11,25,26,27,28,29,30,31,32,33,34,35,36,37,38,39]. Three different crystal polymorphs have been reported, i.e., α (triclinic), α’, and β (monoclinic) structures [27,29,32,34]. In contrast to PET, which preferably crystallizes in the α-form [35,40], the β-form is the most common one for PEF [9,27,29,32,34]. Diffraction peaks have been reported to occur at *2ϴ =* 15.8°, 20.2°, 25.2°, and 27.2° [9,29,32,35] and *2ϴ =* 16.2°, 18.0°, 20.8° ([10]), 23.3°, and 26.7° [8,10,11,31,36,37], respectively. Bulk crystallization was reported to result in α-crystallization [32]; melt-quenched PEF was reported to yield α’-PEF, while the β-form was obtained from slow solvent crystallization [27,39]. An additional XRD-signal was reported to occur at *2ϴ* = 19.3 [29] and was attributable to the α-structure and to the similar α’-structure. However, no stretching conditions have been considered at all. Forestier et al. [28] obtained a monoclinic structure by strain-induced crystallization, as is the case with static crystallization. In the diffractograms of Koltsakidou et al., single signals for β-PEF were reported to show up at *2ϴ* = 37.5° [9].

Standard melting enthalpy $\left(\Delta H_{m}^{0}\right)$
values for PEF in the *β*-form of 137–140 J·g^−1^ [9,36], comparable to those of PET [41], have been reported. Van Berkel [33] and Codou et al. [42] proposed higher values of 185 and 187 J·g^−1^, respectively. Degrees of crystallinity of 25 and 33% for annealing and melt crystallization, respectively, have been reported [9,36,38,42], reaching 50% after solid state polycondensation (SSP) [13]. The coexistence of *α*- and *β-*crystal form (multiple melting endotherms), comparable to PET, was demonstrated for purified PEF by Righetti et al. [34], as well as by Zheng and Pan [43], and was found to be correlated to catalysts acting as nucleation agents for the α-form.

In order to be suitable for (multi-) filament spinning and, especially, for the production of high-strength yarns, it is mandatory to reach an intrinsic viscosity ([η]) of at least 0.6 dL·g^−1^, as is the case with fiber-grade PET [44]. This requires the provision of monomers of sufficient quality and purity for the polycondensation of PEF without by-products, which also avoids colorations [39,45,46]. In addition, a high
[η] value influences *T_g_* [47].

In principle, PEF can be prepared via two routes. The first entails direct synthesis using dimethyl 2,5-furandicarboxylate (FDME), and the second is based on the transesterification of FDCA. While the FDME route results in higher purity PEF, many industrial plants for PET (also usable for PEF) are no longer equipped for the removal of the methanol that is released during the process, restricting synthesis to the more challenging FDCA route [39,48].

FDCA can be purified by crystallization [49]. The state-of-the-art production of FDCA entails the hydrothermal conversion of fructose into hydroxymethyl furfural, which can be further oxidized to FDCA [50,51]. The latter step can be efficiently accomplished in a biocatalytical manner [52] with different fermentation protocols [53,54,55,56,57]. Complementary, purely synthetic protocols [58] and chemo-enzymatic cascades [59] exist.

The aim of this work was to overcome the
[η]-limitation of PEF using both the FDCA and FDME routes to produce fiber-grade PEF using SSP. PEF prepared via the FDME route served as the reference material. The most promising PEF batches were used to produce multifilament yarns, which were also drawn to obtain FDY with high strength (>50 cN·tex^−1^). XRD and DSC-measurements were carried out along all process steps—those encompassing synthesis, SSP, LOY/POY, FDY, and recycled granules—to study crystallization behavior and to retrieve information about the yarn-mechanics and storability of LOY.

## 2. Materials and Methods 

### 2.1. Chemicals

MEG was purchased from Brenntag GmbH (Essen, Germany). Titanium (IV) tetra(*n*-butoxide) (≥99%, Alfa Aesar) was purchased from VWR International LLC.; triphenyl phosphate (p.a.) was purchased from Merck KGaA (Darmstadt, Germany); antimony (III) oxide (99%+) was purchased from Acros Organics (Geel, Belgium) and used as received. FDME (>99.5%,) was obtained from AVA Biochem BSL AG (Muttenz, Switzerland) and was used as received. FDCA (>99%) was purchased from either Holypharm Biotech Co., LTD. (Hangzhou, China) or Purac Biochem (Gorinchem, The Netherlands) and used as received.

### 2.2. General Procedure for PEF Synthesis 

For PEF synthesis, a 20 L steel autoclave (Juchheim GmbH, Bernkastel Kues, Germany) with a pressure stability rating of 15 bar was charged with FDME or FDCA (1 eq), MEG (2.2 eq), and titanium (IV) tetra(n-butoxide) (225 ppm with respect to FDME/FDCA). The reactor was flushed with nitrogen and then sealed and heated to 250 °C (FDME)/200 °C (FDCA) for 4–7 h while maintaining the temperature at atmospheric pressure. The methanol and water that were liberated during the reaction were removed from the reaction product via an ascending column and a downstream descending condenser. A reduced pressure of less than 10 mbar was applied for several hours at 260 °C for the complete removal of excess methanol/water. The progress of the polymerization was monitored via the torque of the stirrer, which reached a maximum value of 31 Nm. The discharge of the polymer from the reactor was accomplished in the form of a strand, which solidified directly after the discharge valve in an ice bath. This strand was further processed in a pelletizer to form extruded pellets. The pellets were dried for further processing under reduced pressure. A figure of the melt release from the reactor is given in Appendix A.

### 2.3. Solid-State Polycondensation

PEF, as synthesized, was further treated in a vacuum oven for SSP. A sub-atmospheric pressure of <10 mbar was applied. The PEF pellets were stored for 1 day to 160 °C; then, heat treatment was continued for another 1–3 weeks at 180 °C, as shown in Appendix A.

### 2.4. Intrinsic Viscosity Measurements

The solution viscosity of the samples was measured at 25 °C with an Ubbelohde-Ia viscometer in dichloroacetic acid (99%) according to DIN EN ISO 1628–5 for PET.

### 2.5. Carboxyl End Group (CEG) Determination

The carboxyl end-group content (CEG) of the PEF polyesters was determined according to ASTM D7409–15 via the potentiometric titration of a solution of the polyester in a *m*-cresol/dichloro-methane mixture. A KOH solution in isopropanol was used as a standard solution. The average of two independent measurements was used for each sample. CEG measurements were solely performed for PEF that was not subjected to SSP, because after SSP, the samples were no longer soluble in the solvent used for the measurement due to their high crystallinity and higher molar masses.

### 2.6. Size Exclusion Chromatography (SEC) 

An Agilent Technologies 1260 Infinity II High Temperature GPC System (GPC 220, Agilent Technologies, Inc, Santa Clara, USA) equipped with a refractive index detector was used and operated at 50 °C in *m*-cresol as eluent. Twenty milligrams of a PEF sample were dissolved in a 20 mL *m*-cresol solution at 80–120 °C for 0.5–3 hours. Three consecutive PLgel Olexis columns (0.013 Å pore size) and one precolumn were used while applying a flow rate of 0.4 mL·min^−1^. For the recording and evaluation of the chromatograms, the GPC/SEC (size exclusion chromatography) software of Agilent Technologies (Santa Clara, USA) was used. Narrow distributed polystyrene standards with molar masses from 1681 to 2,000,000 g·mol^−1^ were used for calibration.

### 2.7. Determination of the Moisture Content

The residual water content (after SSP) was determined by Karl Fischer titration, which was performed on an “899 Coulometer” and an “885 Compact Oven SC” (both: Deutsche METROHM GmbH & Co. KG, Filderstadt, Germany) at 140 °C. The resulting water content was <150 ppm.

### 2.8. Shear Rheological Characterization

Shear rheological experiments in the temperature- and time-sweep modes were performed on a “Physica MCR 501” rheometer (Anton Paar Group AG, Graz, Austria) in plate–plate geometry at different temperatures. Polymer granules were placed on the lower plate (25 mm in diameter), and the gap was adjusted to 1.0 mm. Afterwards, excess material was removed and the test was run under nitrogen atmosphere (strain: 1%, shear rate: 1 s^−1^). The strain amplitude had previously been proven to be in the linear viscoelastic regime by strain sweep tests at a constant shear rate of 1 s^−1^. 

### 2.9. Melt Spinning

Filament yarns were melt-spun using a single screw extruder (Ø 25 mm × 20 D) with a mixing section from Extrudex GmbH (Mühlacker, Germany) and wound on an ASW Barmag winder (Barmag Saurer GmbH & Co. KG, Remscheid, Germany). Filaments were spun with an injection speed of 3.06 m·min^−1^ on four spinning positions with a 40-filament die package (circular spinnerets with D = 500 μm; L = 1000 μm; and a staggered orifice arrangement), passed through a quench duct (rectangular:, = 40 × 45 cm, length = 100 cm; one sided air-flow; air speed of 0.48 m·s^−1^; 50.1% relative humidity; and 23.6 °C). For take-up, a pair of godets (D = 125 mm) was used in S-wrap. A spinning draught was applied by choosing godet speeds 25 m·s^−1^ (First) and 50 m·s^−1^ (Second) slower than the winder. “Limanol ST9” 16% in demineralized H_2_O (Schill + Seilacher, Böblingen, Germany) was used as spin finish.

### 2.10. Drawing of As-Spun Yarns

The as-spun yarns (LOY/POY) were drawn on a test stand with heatable drawing rollers and static heaters. Figure 1 schematically shows the basic arrangement of the godets that defines the drawing zones per-drawing, main-drawing, post-drawing, interlacing, and winding, as well as the temperature range applied in the drawing trials. For pre-drawing, the draw ratios (DRs) were 1.00–1.01, and for post-drawing, the DRs were 0.90–1.10. The relevant main draw ratios lie at values around 1.9 for POY and 2.7 for LOY. In addition to heatable godets, there was a hot plate in the main drawing zone to support the drawing process by heating the yarn above *T_g_*. In the post drawing zone, the heatable godets or non-contact heaters allowed for additional heat treatments of the yarn, e.g., for shrinkage reduction. After drawing, the multifilament yarns were interlaced with an air jet (LD22.02 of Temco Textilmaschinenkomponenten GmbH, Hammelburg, Germany; air pressure = 2.0 bar) to obtain a closed yarn structure for downstream processes.

### 2.11. Testing of Yarn Titer

The titer of the yarns was measured using a yarn reel and a fine scale according to DIN EN ISO 2060.

Additionally, the uniformity of yarns was briefly measured with a capacitive measuring method on an Uster-Tester 5 according to DIN 53817 T2. The coefficient of variation was characterized to lie around 2.1–2.2%.

### 2.12. Tensile Testing

The tensile strength (σ_m_) and the elongation at break (ε_B_) of the yarns, as well as the elastic modulus (E) as secant modulus between 0 and 1% elongation, were determined with the aid of an Uster TensoRapid (Uster Technologies AG, Uster, Switzerland) according to DIN EN ISO 2062. Twenty measurements per sample were carried out while applying a test speed of 500 mm·min^−1^ and a sample length of 500 mm using a preload of 0.5 cN·tex^−1^.

### 2.13. Differential Scanning Calorimetry (DSC)

DSC (differential scanning calorimetry) measurements were carried out under air (20 mL·min^−1^) on a Q2000 differential scanning calorimeter (TA Instruments Inc., New Castle, DE, USA) while applying a heating rate of 10 K·min^–1^. The sample mass was 2 mg. The melt enthalpy Δ*H_m_* and melting peak temperature *T_m,p_* were determined from the heat flow–temperature curves, as well as the glass-transition temperature *T_g_*. One measurement per sample was carried out using the 1st heating cycle only, for both the granules and the fiber samples. This is because the second heating cycle gives no information on the history of the material and because the (re-)crystallization of PEF is very slow and therefore not displayable by DSC measurements.

The degree of crystallinity *χ_c_* was calculated by standardizing the melt enthalpy to the standard melt enthalpy Δ*H_m_*_,0_, as shown by Equation (1).
(1)$$\chi_{c}\;=\;\frac{\Delta Hm}{\Delta Hm,0}$$

The standard melt enthalpy Δ*H_m_*_,0_ used was 137 J·g^−1^ according to the literature [9].

### 2.14. X-ray Diffraction 

XRD measurements were recorded on a D/Max Rapid II diffractometer (Rigaku Corp, Akishima, Japan) using monochromatic Cu *Kα* radiation (*λ* = 0.15406 nm; *U_acc_* = 40 V; *I_acc_* = 30 mA) and an image plate detector. A scanning rate of 0.2° min^−1^ and a step size of 0.1° were applied. The measurement time was 1 h for all investigated samples. The diffraction patterns were analyzed using the PDXL 2 software, and pseudo-Voigt profile fitting was chosen for the evaluation of reflex positions and crystalline fraction determination. The degree of crystallinity *χ_c_* was calculated according to Equation (2),
(2)$$\chi_{c}\;=\;\frac{\sum I_{c}}{\sum \left(I_{c}+I_{a}\right)}$$
where *I_c_* and *I_a_* are the integrated intensities of crystalline reflexes and amorphous reflexes, respectively.

The samples were prepared as follows: granules were milled and pressed, and then filaments were bunched and arranged parallel on the carrier.

### 2.15. Filament Recycling

A 20 L steel autoclave (Juchheim GmbH, Bernkastel Kues, Germany) was charged with PEF fibers, which were washed at 60 °C to remove the spin finish and short cut to 0.5 cm. The reactor was flushed with nitrogen and then sealed and heated to 250 °C for 30 min while maintaining atmospheric pressure and temperature. The progress of the melting was monitored via the torque of the stirrer, which reached a maximum value. The discharge of the polymer from the reactor was accomplished in the form of a strand, which solidified directly after the discharge valve in an ice bath. This strand was further processed in a pelletizer to form extruded pellets. The pellets were dried for further processing under a reduced pressure.

## 3. Results and Discussion

### 3.1. PEF Synthesis

The synthesis of PEF was achieved via the esterification of FDCA and MEG (Scheme 1a) and the transesterification of FDME with MEG (Scheme 1b), followed by polycondensation in the melt.

For the FDME route, several catalyst systems that are commonly used in PET synthesis were tested. The molar mass of the produced polymers was then increased by consecutive SSP.

Syntheses were accomplished on a 10 kg scale. Titanium (IV) tetra(*n*-butoxide) turned out to be among the most effective tested catalysts in terms of reaction time and high [*η*] value; however, antimony (III) oxide also showed good results at prolonged reaction times and by applying SSPs. Finally, all experiments were carried out with titanium (IV) tetra(*n*-butoxide) (Table 1).

With titanium (IV) tetra(*n*-butoxide) as the catalyst, [*η*] values up to 0.600 dL·g^−1^ were achieved at optimized reaction temperatures with both FDME and FDCA. [*η*] was then further increased by SSP using the SSP parameters of PET adjusted to the different *T_g_* and *T_m_* values of PEF. After successful optimization, [*η*] values > 0.790 dLg^−1^ (PEF-05 and PEF-08) were achieved. These values correspond to those of technical textiles for PET. While PEF was clear and colorless prior to SSP, it became turbid after SSP, indicating post-crystallization. SEC confirmed the increase in *M_n_* with increasing [*η*]. Polydispersities (*Đ*) were broader after SSP but in a typical range. An analysis by SEC have lower values of molar masses for PEF-05 and PEF-06, while the Ubbelohde analysis gave higher values of [*η*] compared to PEF-03. This might have been due to degradation of PEF-05 and PEF-06 during the SSP-process after an increase of the molar masses in the first place. The degradation was better detected by SEC than by the viscosity measurements.

The FDCA route can be expected to be the preferred PEF synthesis route in the industry due to the similarity to the TPA route. However, suitable FDCA purities have only recently been developed (PEF-08).; therefore, commercial FDCA-based PEF samples (PEF-07 and PEF-08) were synthesized as well. PEF-08 features higher molar masses, and a further significant increase was noted after SSP compared to the FDCA-based PEF (PEF-07) from other sources.

The CEG values of PEF samples of the FDME route were about 50 µeq·g^−1^ and thus comparable to those for commercial PET. They also explained the high [*η*] values after SSP (PEF-01–PEF-06), while very low CEG values were measured for PEF-07. This indicated that the number of carboxylic end groups was low (7 µeq·g^−1^) compared to the number of hydroxy end groups, and a linkage was most unlikely and resulted in poor [*η*] values after SSP. By contrast, PEF-08 had a sufficient CEG value of 21 µeq·g^−1^, thus allowing for an [*η*] of 0.850 dl·g^−1^ after SSP.

None of the PEF samples showed turbidity after polycondensation, thus suggesting slow crystallization. In order to identify any slow, continuous crystallization of the spun fibers that could impede conversion into FDYs at a later stage, DSC and XRD measurements were carried out on selected samples.

The DSC curves of the PEF granules showed no exotherms or endotherms, but they did show *T_g_* values of 81.4 and 80.1 °C for PEF-3 and PEF-4 before SSP, respectively. No melting peak was detected in the heating curves. After SSP, distinct melting peaks occurred in the heat curves, as summarized in Table 2, complemented by the degree of crystallinity *χ_c_* obtained by XRD. *T_g_* was significantly increased for PEF-04 but not for PEF-03.

The crystallinity values of PEF-03 and PEF-04, calculated from the standard melting enthalpy of 137°J·g^−1^ [9], were 49% and 46%, respectively, independent of the intrinsic viscosity reached. The crystallinity values were in line with the highest values obtained by Chebbi et al. [13] for PEF after SSP when using different catalysts. While the melt enthalpy and the peak temperature lay in the same range, an increase in *T_g_* was observed for the PEF-04 sample with an [*η*] of 0.78 dL·g^−1^. Overall, the XRD results agreed well with the DSC results.

### 3.2. Spinning of PEF

Both LOYs and POYs were spun from selected PEF lots. Table 3 shows the respective titers and the corresponding thermal characteristics from the first DSC heat run (see Section 2.13). The DSC curves of all samples tested in this study are given Appendix A.

The *T_g_* values of all yarns were close to 90 °C. Unsurprisingly, the LOY showed no or low crystallinity, while the POYs had a less pronounced melting peak (~30 J·g^−1^) with a lower melting peak temperature than the granules. LOY-03 showed no sign of pre-crystallization when stored below 0 °C (a→b). The properties of FDY-P-03-05 and FDY-L-04 are shown in Table 4.

The melting enthalpies of the FDYs were in the range of 41–49 J·g^−1^ and thus lower than those of the SSP-treated granules. The melting temperatures were in the same range than those of the LOYs and POYs but lower than those of the granules. In line with a lower degree of crystallinity, a significantly lower strength of the yarn was reached with LOYs. Additionally, a reduced strength was obtained for FDY-P-04, which was a consequence of the spinning temperature being kept equal to that of FDY-P-03 but the [*η*] being in the range of PEF-05 (see Table 1 and Table 3). The corresponding shear-rheological characterization is given in Appendix A. A high tensile strength of >65 cN·tex^−1^ and a high Young’s modulus of >1500 cN·tex^−1^ showed that PEF was competitive with PET and suitable for the production of technical yarns like tire cords [60,61]. Still, the elongation values were somewhat low and need to be enhanced [62].

### 3.3. Characterization of the Crystal Structure of PEF Along the Entire Yarn “Life-Cycle”

Figure 2 shows the X-ray diffraction images, which significantly differed for each different process step. The transformations from amorphous to crystalline phases by SSP treatment, as well as the conversion to anisotropic fibers and crystalline FDY, are clearly visible. The corresponding diffraction patterns of the previously discussed PEF-samples are given in Figure 3.

Figure 3a shows the diffraction patterns of the milled granule samples after SSP. Several crystalline reflections occurred at *2θ* = 16.2° ((101)), 17.9° ((004)), 19.4° ((1$\bar{1}$0)), 20.6° ((103)), 23.5° ((110)), and 26.9° ((020)) in accordance with the literature [8,10,11,31,36,37,63]. The present crystalline phase can be referred to the α-phase, as the reflex at 19.4° can be taken as typical marker that is absent in the α’- and β-phases [30]. This reflex was present in all three granules, but it was very little for the FDCA-derived PEF-08 sample, meaning there was probably a mixture of the α- and α´-phases. As no reflex at *2θ* = 9.5° was observed, the presence of the β-phase could be excluded. In addition to the crystalline fraction, an amorphous phase (*2θ* ~ 20°) was present and quantified by the peak deconvolution method (see Appendix A). As stated in Table 2, a crystallinity of 43% was determined for PEF-03 and PEF-04, both synthesized from FDME. By contrast, PEF-08, synthesized via the FDCA route, showed a slightly higher crystallinity of 47%.

While LOY and POY were completely amorphous, as shown in Figure 3b, the obtained FDY yarns (Figure 3c) were semi-crystalline and anisotropic fibers. The POY-derived FDY showed several crystalline reflexes at *2θ* = 8.8°, 16.6°, 20.8°, 26.3°, and 42.7°, corresponding the (002), (101), (103), (020), and (0010) lattice planes, respectively (for more detail, see Appendix A). The crystalline reflexes could be assigned to the α- and/or the α´-phase, as no clear peak at 19.5° could be identified for these semi crystalline fibers. In case of the LOY-derived FDY, these signals were even less pronounced, thus indicating a dependence on the drawing ratio and PEF phase formation, as well as a lower degree of crystallinity. Recently, Forestier et al. discussed a similar signal at *2θ* ≈ 43°, occurring on uniaxial stretched PET during mechanical testing, that intensified with the stretching ratio due to a denser chain packaging along the stretching direction and that corresponded to [10] lattice plane. [28,63] These meridional reflexes are also clearly visible for all POY-based FDY samples in the diffraction image (Figure 2e) and the meridional diffraction pattern (see Figure 4). FDY drawn from LOY showed an additional meridional reflex at ~35.3° that belonged to the (008) lattice plane [64].

Additional lattice planes became visible via the negligence of the equatorial lattice planes and are shown in Appendix A Further, a transformation from an amorphous signal to the formation of three indistinct peaks at winding speeds >4000 m·min^−1^ was reported due to orientation-induced crystallization. These peaks intensified with the increasing winding speed along the c-axis. Comparable findings were also reported for PEF samples that were stretched uniaxially via mechanical testing [63]. In accordance to our findings on stretched filaments, the authors observed that the previously amorphous signal changed to the formation of transversal diffractions (two-to-three peaks in the hk0 planes), which increased in intensity and discreteness with an increasing stretching ratio [63].

Prior to SSP, the polymer showed a completely amorphous signal according to DSC, similar to the structure of the as-spun yarns in XRD. This is illustrated in Figure 3d. The recycled material (red curve) again showed an amorphous signal because it was fully molten, quenched in ice-water, and pelletized.

Considering the regime of *2θ* > 30°, peaks became visible for the pelletized samples (post-SSP) at *2θ* = 32.6, 35.2°, 39.2°, 42.9°, 47.6°, and 50.1°, while LOY and pre-SSP pellets exhibited an amorphous signal in this range. In this range, Forestier reported meridional diffractions after the uniaxial stretching of PEF [28,63]. The discrete signal of FDY at *2θ* = 42.7° was discussed before.

## 4. Conclusions

PEF was synthesized on a multi-kg-scale via the FDCA route. SSP successfully increased the intrinsic viscosity, thus rendering the final PEF usable for melt spinning. It was shown that both the FDME and the FDCA routes are capable of producing PEFs with high intrinsic viscosities, suitable CEG values for further processing by SSP, and molar masses and molar mass distributions for the successful melt spinning into technical textiles. Multifilament yarns with tensile strengths >65 cN·tex^−1^ were successfully spun and drawn on the pilot scale. Elongation at break (<10 %) still has to be further improved to be suitable for applications of comparable PET yarns. The crystalline structure of the synthesized PEF was determined by XRD and revealed an α-form. By processing LOY to FDY, a transformation from an amorphous to a “semi-crystalline” signal was observed, referable to strain-induced crystallization. Signals in the range of *2θ* > 30° also allowed for the distinguishing between the different process steps.

In summary, the FDME route leads to PEF-based melt-spun fibers with exceptional mechanical properties. Complementary, the FDCA route has been further developed to obtain PEF with similar properties.

## Data Availability

Data available on request due to privacy restrictions. The data presented in this study are available on request from the corresponding author. The data are not publicly available due to running project issues.

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
