# Peer review of "Poly(Ethylene Furanoate) along Its Life-Cycle from a Polycondensation Approach to High-Performance Yarn and Its Recyclate"

_materials, 2021, doi:10.3390/ma14041044_

Round 1

Reviewer 1 Report

This paper focuses on the polycondensation of PEF for high-performance yarn application. The paper was well organized and conducted out adequate experiments to verify their hypotheses. the discussion was accurate and concise. Two points may need the authors to further address.

  1. The authors mentioned PEF is a biobased polymer, is it bidegradrable or can it be commercial composted or home composted? The authors may address this in the instruction if the authors claim it is a bioplastic.
  2. what are PEF-01 to PEF-08 in table 1? The authors should further describe how to get this nomenclature.

Author Response

Review 1:

This paper focuses on the polycondensation of PEF for high-performance yarn application. The paper was well organized and conducted out adequate experiments to verify their hypotheses. the discussion was accurate and concise. Two points may need the authors to further address.

 We thank the reviewer for the suggestions, which will help to get the manuscript clearer for the readers understanding.

1.The authors mentioned PEF is a biobased polymer, is it bidegradrable or can it be commercial composted or home composted? The authors may address this in the instruction if the authors claim it is a bioplastic.

PEF is bio-based, but not biodegradable for the state of the art. We will address this with a annotation in the introduction on p. 1, line 38.: “Although PEF is not biodegradable for the current state of research, it is fully recyclable [7] […]”

2. What are PEF-01 to PEF-08 in table 1? The authors should further describe how to get this nomenclature.

The sample-nomenclature refers to the experiment order within the project. This will be added to the table caption (p.6, line 238).

Reviewer 2 Report

The waste plastic problem is becoming a global issue, and the development of alternative materials is urgently required. PEF is gathering recent attention as an ecofriendly bioplastics of faster decomposing. Thus, the practical research introduced in the manuscript is valuable for readers. However several information about the processing is missing in the manuscript, and thus the minor modification on the current manuscript would make the manuscript more valuable for the readers.

  1. In the first sentence of the abstract, author stated that their experimental scale as ‘the semi-industrial-scale’. However the current scale of 20 liter autoclave reactor may not be referred as 'semi-industrial’scale, and the scale of melt-spinning of 40 orifices is barely over lab-scale. Thus I suggest author to revise the expression of their experimental scale from ‘semi-industrial’ down to the bench or semi-pilot to prevent confusion of your readers.

Compared to the PEF synthesis session in the manuscript, many detailed information about the melt-spinning process are missing; spinneret structure (circular or rectangular), orifice arrangement (parallel or staggered?), quenching environment (active quenching or natural cooling?), and spinline information. Also, it would be nice to provide fiber diameter deviation.

Author Response

First of all, the authors would like to thank the very helpful reviews. In this cover letter, we will address the changes made in the revision of the manuscript.

Reviewer 2: The waste plastic problem is becoming a global issue, and the development of alternative materials is urgently required. PEF is gathering recent attention as an ecofriendly bioplastics of faster decomposing. Thus, the practical research introduced in the manuscript is valuable for readers. However several information about the processing is missing in the manuscript, and thus the minor modification on the current manuscript would make the manuscript more valuable for the readers.

We thank the reviewer for the help to improve our manuscript in order to make our work more comprehensible and reduce unclarities for readers.

  1. In the first sentence of the abstract, author stated that their experimental scale as ‘the semi-industrial-scale’. However the current scale of 20 liter autoclave reactor may not be referred as 'semi-industrial’scale, and the scale of melt-spinning of 40 orifices is barely over lab-scale. Thus I suggest author to revise the expression of their experimental scale from ‘semi-industrial’ down to the bench or semi-pilot to prevent confusion of your readers.

We will change “semi-industrial-scale” to “pilot-scale” and “multi-kg-scale” respectively: p.1, line:13, p.10, line: 351, p. 10, line:356)

Compared to the PEF synthesis session in the manuscript, many detailed information about the melt-spinning process are missing; spinneret structure (circular or rectangular), orifice arrangement (parallel or staggered?), quenching environment (active quenching or natural cooling?), and spinline information. Also, it would be nice to provide fiber diameter deviation.

We accomplished the information about the melt spinning process in paragraph 2.9 (p.4, line: 156f) with further spinline information:

 Circular spinneret structure

  • Staggered orifice arrangement
  • Active quenching with air in a rectangular quench duct (one sided air-flow/supply, 40x45x100cm)
  • Godet diameter: mm

 Also, a paragraph addressing the diameter derivation was added to 2.11 (p.5., line: 180): „Additionally, the uniformity of yarns was measured briefly with a capacitive measuring method on an Uster-Tester 5 according to DIN 53817 T2. The coefficient of variation was characterized to lie around 2.1 - 2.2% respectively.”  
